# Production of Vitamin K by Wild-Type and Engineered Microorganisms

**DOI:** 10.3390/microorganisms10030554

**Published:** 2022-03-03

**Authors:** Min-Ji Kang, Kwang-Rim Baek, Ye-Rim Lee, Geun-Hyung Kim, Seung-Oh Seo

**Affiliations:** Department of Food Science and Nutrition, The Catholic University of Korea, Bucheon 14662, Korea; escape97@catholic.ac.kr (M.-J.K.); rimmy@catholic.ac.kr (K.-R.B.); yerim61@catholic.ac.kr (Y.-R.L.); ruby7216@catholic.ac.kr (G.-H.K.)

**Keywords:** vitamin K, production, microorganisms, fermentation, metabolic engineering

## Abstract

Vitamin K is a fat-soluble vitamin that mainly exists as phylloquinone or menaquinone in nature. Vitamin K plays an important role in blood clotting and bone health in humans. For use as a nutraceutical, vitamin K is produced by natural extraction, chemical synthesis, and microbial fermentation. Natural extraction and chemical synthesis methods for vitamin K production have limitations, such as low yield of products and environmental concerns. Microbial fermentation is a more sustainable process for industrial production of natural vitamin K than two other methods. Recent advanced genetic technology facilitates industrial production of vitamin K by increasing the yield and productivity of microbial host strains. This review covers (i) general information about vitamin K and microbial host, (ii) current titers of vitamin K produced by wild-type microorganisms, and (iii) vitamin K production by engineered microorganisms, including the details of strain engineering strategies. Finally, current limitations and future directions for microbial production of vitamin K are also discussed.

## 1. Introduction

Vitamin K refers to fat-soluble vitamins which play a role in human health, including blood coagulation and bone health [1]. Vitamin K exists as phylloquinone or menaquinone in nature [2]. Phylloquinone is called vitamin K1 and is found in the green leaves of vegetables [3]. Menaquinone is called vitamin K2 and is mainly produced by bacteria [4]. Various fermented foods made by bacteria such as *Natto* (Japanese fermented soybean), *Cheonggukjang* (Korean fermented soybean), and various cheeses are the main sources of menaquinone uptake [5,6]. Menaquinone is also found in foods of animal origin, such as eggs and meat, in small amounts [6]. Vitamin K has the 2-methyl-1,4-naphthoquinone ring as a core structure, and the side-chain at the C3-position of the naphthoquinone ring structure (Figure 1A) [5]. Phylloquinone has a partially unsaturated side chain consisting of one isopentenyl followed by three isopentyl units, while menaquinones have a fully unsaturated side chain composed of 2 to 13 isopentenyl units [7]. Menaquinone has various subtypes that have a different number (n) of isoprenoid units called menaquinone-n (MK-n) [8]. For example, MK-4 contains four isoprenoid units at the side chain connected to a 2-methyl-1,4-naphthoquinone ring. The structural difference of vitamin K affects biological functions [9]. The isoprene side chain of vitamin K affects half-life and intestinal absorption of vitamin K [10]. Due to the structural difference, biological activity among various vitamin K forms varies despite similar modes of action [11].

In mammals, vitamin K is an essential micronutrient exhibiting several biological functions such as blood coagulation, improving bone health, regulating blood calcium level, preventing cardiovascular disease (CVD), and reducing the risk of cancer (Figure 1B) [12]. Vitamin K is an essential cofactor for the production of blood clotting factors [13,14,15]. Due to this effect, the name of vitamin K came from the German word “Koagulations” [16]. Many clinical studies also suggest that vitamin K plays an important role in bone health, such as preventing osteoporosis [17,18]. Vitamin K has been considered as a powerful micronutrient in aging and age-related diseases [19]. Notably, menaquinone is considered and used as an emerging anti-osteoporosis drug [20,21]. Moreover, vitamin K might be related to reducing risk of CVD by preventing vascular calcification [22,23,24,25]. Vitamin K is also related to brain health since it is a cofactor for the activation of two proteins, Gas6 and protein S, that are related to the cognitive process in the brain [26]. Moreover, vitamin K affects the metabolism of sphingolipids which are associated with cellular events in the brain [26]. A reduced form of vitamin K (KH_2_) directly protects phospholipid membranes from peroxidation by reactive oxygen species (ROS), suggesting its antioxidant activity [27]. Particularly, phylloquinone and MK-4 inhibit activation and accumulation of 12-lipoxygenase (12-LOX) enzyme in cultured neurons and oligodendrocytes, thereby preventing cell death caused by oxidative stress [28,29].

As a nutraceutical and dietary supplement, vitamin K is currently produced by natural extraction, chemical synthesis, and microbial fermentation (Figure 1B) [30]. Several manufacturers of vitamin K mainly use a chemical synthesis method [30,31,32]. Chemical synthesis requires multiple reaction steps, thereby reducing the yield and selectivity of vitamin K by forming unwanted byproducts and isomers [30]. Similar to chemical synthesis, extraction of vitamin K from natural products also has a low yield problem by complex pretreatment processes [33]. Recently, microbial fermentation has been used for industrial production of vitamin K [34]. Microbial production is a more sustainable option than chemical synthesis since it avoids the use of heavy metals, organic solvents, strong acids, and bases [35]. Microbial fermentation can also produce the final products more economically from inexpensive sources of energy, carbon, electrons, and trace nutrients [36]. Even though the chemical synthesized vitamin K can be made for high purity of all-trans vitamin K2 which has biological activity [37,38], it still has product safety concerns for use as a GRAS food supplement [39].

Several wild-type microorganisms isolated from various environments have been used for industrial production of vitamins [39,40]. Native vitamin producers have several limitations, such as low yield and productivity with genetic instability [41]. To produce vitamins with high yield and stability through fermentation, metabolic engineering of microorganisms can be applied [42]. Furthermore, bioprocess engineering approaches such as optimization of fermentation conditions and medium composition can also be applied to improve the yield of products [43]. Various strain engineering methods were applied to vitamin productions by microbial cell factories, such as *Bacillus* and lactic acid bacteria (LAB) [44,45]. Recently, advanced genome editing tools such as the CRISPR/Cas9 system have been used for vitamin production in bacteria such as *E. coli* [46]. In this review, we provide recent progress on vitamin K production by wild-type and engineered microorganisms with details of strain engineering and fermentation strategies. We also discuss current limitations and future directions for microbial production of vitamin K.

## 2. Production of Vitamin K by Wild-Type Microorganisms

### 2.1. Bacillus

*Bacillus subtilis* var. *natto* is a well-known vitamin K producer isolated from *Natto* [47]. *B. subtilis* is a GRAS (generally recognized as safe) host microorganism widely used for the production of food ingredients [48]. Owing to this property, many researchers have attempted to produce vitamin K such as menaquinone using *B. subtilis natto* (Table 1). Berenjian et al. (2011) produced MK-7 using *B. subtilis natto* strain by optimizing the medium composition and fermentation conditions [49]. The authors found optimal fermentation medium for MK-7 production, consisting of 5% (*w*/*v*) yeast extract, 18.9% (*w*/*v*) soy peptone, 5% (*w*/*v*) glycerol, and 0.06% K_2_HPO_4_. After the fermentation of *B. subtilis natto* at 40 °C for 144 h, the maximum titer of MK-7 reached 62.32 mg/L [49]. As a follow-up study, Berenjian et al. (2011) attempted to produce MK-7 in fed-batch fermentation using glycerol by *B. subtilis natto* strain. The authors achieved 68.6 mg/L of MK-7 in 144 h of fermentation by intermittent addition of 2% (*w*/*v*) glycerol during fermentation [50]. Berenjian et al. (2012) performed scale-up experiments for MK-7 production using the same strain of *B. subtilis natto* in a 3 L bioreactor. As a result, the titer of MK-7 increased up to 86.48 mg/L [51]. Berenjian et al. (2014) optimized stirrer speed and aeration rate in a 3 L bioreactor for the production and recovery of MK-7, resulting in increased titer of MK-7 up to 226 mg/L [52]. These results suggested that MK-7 production by wild-type *B. subtilis natto* strain improves by optimization of medium composition and fermentation conditions, followed by scale-up of fermentation. Interestingly, Hu et al. (2017) utilized surfactants to increase MK-7 production by the *B. subtilis natto* R127 [53]. The titer of MK-7 was 40.96 mg/L when 20 g/L of soybean oil was used as a surfactant [53]. Among the surfactants they used, a non-ionic surfactant such as soybean oil was more effective than other ionic surfactants for MK-7 production [53].

Mahdinia et al. used a biofilm reactor with plastic composite supports (PCS) to produce MK-7 by the *B. subtilis natto* [54,55,56,57,58,59]. Using biofilm reactor is a good strategy for enhanced production of MK-7 and reduced lag phase periods with optimization of fermentation conditions [72]. When the fermentation medium containing glucose was optimized, MK-7 produced up to 28.7 mg/L [54]. When the fermentation medium containing glycerol was used, MK-7 titer was 14.7 mg/L [59]. A study performed a static fermentation using McCartney bottles, resulting in increased titer of MK-7 to 32.5 mg/L [60]. In the MK-7 production modeling study, the authors predicted and confirmed that MK-7 production by the wild-type *B. subtilis natto* exhibited a mixed-metabolite pattern [73]. Donya Novin et al. (2020) produced MK-7 from a newly isolated *B. subtilis natto* using the medium containing milk in a bioreactor [61]. The maximum titer of MK-7 reached 3.54 mg/L through optimization of agitation and aeration as the key operating conditions [61].

Other than *B. subtilis natto* strain*, B. subtilis* NCIM 2708 was used to produce MK-7 by solid-state fermentation [62]. When the medium composition and fermentation condition were optimized, the titer of MK-7 led to 39.039 mg/L for 24 h of fermentation [62]. Another approach to increase MK-7 production by *B. subtilis* was using iron oxide nanoparticles coated with 3-aminopropultriethoxysilane (IONs@APTES), which might enhance secretion of MK-7 out of the *Bacillus* cells [63]. *B. subtilis* ATCC 6633 was utilized to produce MK-7 in the medium containing 200 mg/mL IONs@APTES, resulting in the maximum titer of MK-7 up to 37.36 mg/L [63].

Other *Bacillus* strains were also used for menaquinone production. One of the examples was *B. amyloliquefaciens* KCTC 11712BP used for manufacturing *Cheonggukjang* with high concentration of menaquinone [64]. Using 4% glycerol supplemented to *Cheonggukjang*, the titer of total menaquinone in *Cheonggukjang* was 12.47 mg/g (0.76 mg/g of MK-4 and 11.71 mg/g of MK-7) [64]. *B. velezensis* ND was also tested for MK-7 production by optimization of fermentation conditions with nitrogen source [65]. The MK-7 production varied with fermentation processes, resulting in 52.9 mg/L by liquid-state fermentation, 73.3 mg/L by biofilm fermentation, and 150.02 mg/kg by solid-state fermentation [65]. These results suggested that wild-type *Bacillus* strains naturally produce vitamin K at a high concentration. The highest titer of MK-7 by the wild-type *Bacillus* strain was 226 mg/L after 100 h of fermentation [52]. Vitamin K production can be improved by bioprocessing engineering, such as optimization of fermentation conditions. Some of the *Bacillus* strains can be utilized for industrial production of vitamin K through large-scale fermentation.

### 2.2. Lactic Acid Bacteria

Multiple forms of menaquinone have been found in fermented dairy foods, which may be produced by lactic acid bacteria (LAB) (Table 1) [74]. In cheese fermentation, *Lactoccocus* ssp. and propionibacteria are known to produce menaquinones [75,76]. Food-related LAB, as well as members of human gut microbiota such as *Bifidobacterium*, can synthesize and provide vitamin K to the human body [71,77].

Morishita et al. (1999) quantified menaquinones produced by LAB, including *Lactococcus* ssp., *Leuconostoc* ssp., *Enterococcus faecalis*, *Lactobacillus* ssp., *Streptococcus* ssp., and *Bifidobacterium* ssp. [66]. The authors conducted fermentations using both growth medium and soymilk. The batch fermentations of LAB were performed with the Rogosa medium containing 20 g/L glucose at 30 °C for 48 h. The menaquinone was extracted from lyophilized cells from the growth medium and from soymilk culture using a mixture of chloroform and methanol. Among the LAB strains examined in this study, *Lactococcus lactis* subsp. *cremoris, Lactococcus lactis* subsp. *lactis*, and *Leuconostoc lactis* were selected as high producers of menaquinone [66]. Menaquinone content of *L. cremoris* YIT 2011 was total 534 nmol/g lyophilized cells, consisting of MK-7 to MK-9. The highest menaquinone content was 717 nmol/g lyophilized cells of *L. lactis* YIT 2027, consisting of MK-8 to MK-10. In the soymilk fermentation, the highest menaquinone content was 2.60 nmol/g of the soymilk culture by *Leuconostoc lactis* YIT 3001 strain, consisting of MK-7 to MK-10 [66]. Liu et al. (2019) increased the titer of menaquinone by *Lactococcus lactis* subsp. *cremoris* MG1363 by optimizing temperature, carbon source, aeration, and mode of energy [68]. The MG1363 strain produced 90 nmol/L of total menaquinone (MK-5 to MK-10) extracted from the medium by a static fermentation using a growth medium containing glucose at 30 °C for 48 h. When the aerobic fermentation at 120 rpm and 30 °C for 48 h was performed using trehalose as a carbon source, the MG1363 strain achieved a 5.2-fold increase in menaquinone production [68].

Lim et al. (2011) confirmed menaquinone production by *Lactobacillus fermentum* LC272 in different culture media [67]. The MK-4 titer by the LC272 strain was 184.94 µg/L in the modified Rogosa medium, and 63.93 µg/L in reconstituted skim milk after extraction [67]. These results suggested that wild-type LAB strains could produce various menaquinone subtypes with decent titers. The shorter fermentation time was remarkable in menaquinone production by LAB. However, the titers of menaquinones produced by LAB were below those of *Bacillus* strains, when it was considered that the wild-type *B. subtilis natto* strain produced 226 mg/L (346.2 µmol/L) of MK-7 [52]. Other LAB strains can be isolated for higher production titer of vitamin K. Some of the LAB strains may be developed as a probiotic strain producing vitamin K in human gut.

### 2.3. Other Microorganisms

Besides *Bacillus* and LAB, other bacteria also produce vitamin K. *Flavobacterium meningosepticum* is one of the menaquinone-producing bacteria (Table 1) [69]. *F. meningosepticum* is a Gram-negative aerobic bacillus, nonmotile, oxidase-positive bacteria found in soil, freshwater, and saltwater [78]. *F. meningosepticum* is difficult to be engineered because of the lack of biotechnological tools compared to other model microorganisms. Rather than strain engineering, various extraction methods have been applied to increase menaquinone production from *F. meningosepticum*. Fang et al. (2019) used a surfactant and ultrasound during fermentation to increase extracellular menaquinone production [70]. The use of surfactant and ultrasound was used to improve the productivity of target products from biotechnological processes without using chemical agents increasing cell membrane permeability [79]. The authors conducted screening of surfactants that could increase the yield of menaquinone, including Triton-100, sodium dodecyl benzene sulfonate, Tween-80, and polyoxyethyleneoleyl (POE) [70]. When they added 1% POE to the medium, *F. meningosepticum* produced the highest yield of menaquinone, reaching up to 15.95 mg/L. Ultrasound treatment was further treated with POE, resulting in enhanced menaquinone production up to 30.03 ± 1.42 mg/L in an aqueous medium [70]. These results suggested that chemical and physical methods, such as ultrasound and surfactant, might affect cellular permeability by forming pores temporarily in the cell membrane, thereby increasing the secretion of target metabolites out of the cells.

Wei et al. (2018) attempted to overproduce menaquinone through extraction from the cultured cells of *F. meningosepticum* by using various organic solvents. The authors cultured *F. meningosepticum* aerobically in 18 L of medium containing glycerol for 6 days, and harvested cells followed by freeze-drying. The dried cells of *F. meningosepticum* were treated with 5 mL of organic solvents including methanol, absolute ethyl alcohol, acetone, n-butanol, dichloromethane, isopropanol, acetic acid, acetonitrile, n-hexane, and petroleum ether. They finally selected methanol as an efficient organic solvent and produced menaquinone up to 1.88 mg/g DCW from *F. meningosepticum* [69].

Other than *F. meningosepticum*, many efforts have been made for finding a novel microbial strain that produces vitamin K naturally. Cooke et al. (2006) reported vitamin K-producing bacteria isolated from the neonatal fecal flora, including *Enterobacter agglomerans, Serratia marcescens,* and *E. faecium* [71]. The authors confirmed MK–4 production by the newly isolated strains in fermentation medium by LC/MS analysis [71]. Similarly, various intestinal anaerobic bacteria isolated from fecal samples have been evaluated for vitamin K production capacity [80]. Commensal bacteria, including *Bacteriodes* ssp., *Eubacterium lentum*, *Veillonella* ssp., *Wolinella* ssp., *Actinomyces* ssp., *Arachnia propionica*, and *Propionibacterium* ssp., were confirmed as menaquinone producers [81]. These results suggested that a wide variety of bacteria can contribute vitamin K production. However, industrial fermentation of vitamin K requires a decent titer and productivity of the host strains for economical production. Additionally, the host strain may be required to have a GRAS status for production of microbial-derived vitamin K used as a food supplement.

## 3. Production of Vitamin K Using Engineered Microorganisms

### 3.1. Engineered Bacillus

Strain engineering strategies, including random mutagenesis and rational design, can be used to increase the titer, yield, and productivity of target chemicals from fermentation. Various vitamin K-producing *Bacillus* strains have been engineered to increase vitamin production (Table 2). Random mutagenesis using a chemical agent or physical treatment can be first used for the construction of mutant strains exhibiting desirable phenotypes [82]. The well-known chemical mutagen, N-methyl-N’-nitro-N-nitrosoguanidine (NTG), which causes alkylation of guanine or thymine, has been used for the construction of *Bacillus* mutants to overproduce vitamin K [83,84]. Along with the chemical mutagens, various analogs of vitamin K2 precursors such as 1-hydroxy-2-naphthoic acid (HNA) have been treated for generating *Bacillus* mutants with increased metabolic flux to vitamin K biosynthesis [84,85]. The diphenylamine (DPA), which inhibits the biosynthesis of naphthoquinone ring, can also generate the mutant strains overproducing vitamin K [74,77,78]. Sato, T., et al. (2001) used NTG together with DPA to construct the mutant strain of *B. subtilis*. The resulting D200-41 strain produced 19.6 mg/L of vitamin K2 in a 500 mL flask fermentation and 62.1 mg/L of vitamin K2 in static fermentation for 5 days after optimization of carbon and nitrogen source in the growth medium [83]. Similarly, the construction of the *Bacillus* mutant released from feedback inhibition by aromatic amino acids was effective to enhance the vitamin K biosynthetic pathways, since the aromatic amino acids share their biosynthetic pathways with vitamin K [85].

Physical methods for random mutagenesis such as UV and N^+^ ion-beam were also used for generating *Bacillus* mutants with increased vitamin K production [84,85]. Tsukamoto and Kasai et al. (2001) attempted to generate several mutant strains based on the wild-type *B. subtilis* O-2 strain isolated from *Natto* by the UV treatment together with analogs of vitamin K precursors [85]. The resulting mutant *B. subtilis* OUV23481 was utilized for making *Natto* containing vitamin K2 up to 1.719 µg/100 g of *Natto*, which was 1.7 times higher than that made by the parent strain [85]. Similarly, Song, J., et al. (2014) used NTG and HNA with N^+^ ion-beam treatment for construction of the mutant strain based on the wild-type *B. subtilis* BN2-6 strain isolated from *Natto* [84]. The resulting strain BN-P15-11-1 produced 2.5 mg/L of vitamin K2, which was 166% higher than that of the parent strain. Further optimization of the fermentation medium increased vitamin K2 production up to 3.593 mg/L by the BN-P15-11-1 strain [84]. Wang et al. (2018) optimized the nitrogen, carbon, and inorganic salt sources in culture media for increased MK-7 production using the *B. subtilis* BN-P15-11-1 strain [86]. The authors found the optimum composition of medium, which contained soybean curd residue, soya peptone, lactose, and K_2_HPO_4_. As a result, the *B. subtilis* BN-P15-11-1 strain produced about 45 mg/L of MK-7 in the optimized medium in a 3.7 L bioreactor. When the authors optimized oxygen supply conditions using two-stage dissolved oxygen (DO) strategy, the BN-P15-11-1 strain produced 91.25 mg/L of MK-7 in a 3.7 L bioreactor [86]. Puri et al. (2015) constructed the mutant strain based on the *B. subtilis* using 1-naphthol [87]. The 1-naphthol mutant strain produced 12.5 µg/mL of MK-7 in 100 mL flask fermentation for 24 h and 14.4 µg/mL of MK-7 in 100 mL flask fermentation in the presence of Tween 80 for 24 h [87]. These results suggested that vitamin K production by *B. subtilis* strains can be improved by various random mutagenesis strategies using chemical and physical treatments.

Other *Bacillus* strains than *B. subtilis* have been engineered for vitamin K production. Goodman et al. (1976) constructed the menaquinone-deficient mutant strain based on wild-type *Bacillus licheniformis* using kanamycin and shikimate [88]. The resulting strain produced 0.3 nmol/mg of MK-7, which was lower than the titer of MK-7 produced by wild-type strain (0.38 nmol/mg) [88]. Xu, J. Z., and Zhang, W. G. (2017) identified *Bacillus amyloliquefaciens* Y-2 strain from douche, a Chinese salted black bean food [89]. The authors constructed *B. amyloliquefaciens* H.β.D.R.-5 mutant strain based on the wild-type Y-2 strain through multi-round random mutagenesis using HNA, DPA, and β-thienylalanine (βTA) for generating analog resistance. The H.β.D.R.-5 mutant produced 61.3 mg/L of MK-7 in maize meal hydrolysate medium using a 7 L fermenter [89]. Recently, Liu, N. et al. (2021) used the H.β.D.R.-5 mutant to create another *B. amyloliquefaciens* mutant having a high α-amylase activity through adaptive evolution by temperature-induced mutagenesis at a high growth temperature. The resulting heat-resistant mutant MK50-36 produced 57 mg/L of MK-7 in corn starch medium for 6 days of fed-batch fermentation [90].

Along with random mutagenesis, rational strain engineering strategies such as metabolic engineering have been conducted to increase the MK-7 biosynthesis in *Bacillus* [92,93,95,96]. Over the past few years, the MK-7 biosynthesis pathway and many enzymes involved in biosynthesis have been studied, which facilitates metabolic engineering [30]. The biosynthesis pathways of menaquinone in *Bacillus* are categorized into three pathways: the methylerythritol phosphate (MEP) pathway, the shikimic acid (SA) pathway, and the menaquinone pathway (Figure 2). Several studies attempted to overexpress the pathway genes to increase vitamin K production in *Bacillus*. Xu, Yan et al. (2017) performed metabolic engineering study to construct an efficient *B. amyloliquefaciens* platform to produce MK-7 by overexpression of endogenous MK-7 biosynthetic pathway enzymes [91]. After the genetic analysis, six genes involved in the biosynthesis of the naphthoquinone ring were selected for overexpression to increase MK-7 production. The enzymes overexpressed individually in the Y-2 strain were DHNA octaprenyltransferase (MenA), o-succinylbenzoate synthase (MenD), o-succinylbenzoyl-coenzyme A synthetase (MenE), SHCHC synthase (MenH), and heptaprenyl diphosphate synthase (HepS) (Figure 2). Among these enzymes, overexpression of the HepS showed the greatest increase in MK-7 production, up to 273 μg/g DCW [91].

For systematic metabolic engineering, Yang, Cao et al. (2018) developed the modular system for overexpression of the biosynthetic pathway to increase MK-7 production by *B. subtilis* [93]. The authors categorized the MK-7 biosynthetic pathway into four modules: the MK-7 pathway (Module I), the shikimate (SA) pathway (Module II), the methylerythritol phosphate (MEP) pathway (Module III), and the glycerol metabolism pathway (Module IV). They overexpressed key enzymes in each module to increase metabolic flux to MK-7 biosynthesis. The *B. subtilis* MK3 strain overexpressing the *menA* gene in Module I produced 6.6 ± 0.1 mg/L of MK-7, which was a 2.1-fold increase when compared to the parent strain. However, overexpression of the *aroA*, *aroD*, and *aroE* in Module II reduced the MK-7 production in *B. subtilis*. The *dxs*, *dxr*, *yacM*, and *yacN* genes in Module III were overexpressed in the MK3 strain, resulting in increased MK-7 production up to 12.0 ± 0.1 mg/L. Subsequently, the *glpD* gene in Module IV was overexpressed to enhance glycerol utilization in *B. subtilis*, resulting in 13.7 ± 0.2 mg/L of MK-7 production. After the *dhbB* gene was deleted to block the competitive biosynthetic pathway, the final strain of *B. subtilis* produced 69.5 mg/L of MK-7 after 144 h fermentation in a 2 L baffled flask [93]. Similarly, Ma, Y., et al. (2019) overexpressed the rate-limiting enzymes including MenA, Dxs, Dxr, and Idi involved in menaquinone and MEP biosynthetic pathways in *B. subtilis* [92]. The recombinant *B. subtilis* overexpressing the *menA-dxs-dxr-idi* gene cluster showed the highest MK-7 production up to 50 mg/L, which was more than 5 times higher than that of the wild-type strain [92].

Yuan et al. (2020) tried to overproduce MK-4 by *B. subtilis* through combinatorial engineering [94]. The authors overexpressed the *menA*, *menG*, and *crtE* genes from *Synechocystis* sp. PCC 6803 involved in MK-4 synthesis under the strong constitutive promoter P43, resulting in 8.1 mg/L of MK-4. After the knockout of *hepT* gene, which catalyzes the conversion of farnesyl diphosphate to heptaprenyl diphosphate, simultaneous overexpression of *dxs, dxr*, and *ispD*-*ispF* genes in MEP pathway increased the titer of MK-4 up to 78.1 mg/L. Moreover, overexpression of the heterogeneous MVA and menaquinone pathway genes further increased the titer of MK-4 up to 120.1 mg/L in a flask fermentation and 145 mg/L in a 3-L fed-batch fermentation [94].

Interestingly, various quorum sensing (QS) systems have been utilized to increase menaquinone production in *B. subtilis*. The QS-based dynamic regulation has been used for controlling gene expression by cell density changes without using inducers. The authors developed a modular Phr60-Rap60-Spo0A QS system which allows dynamic control of MK-7 synthesis in *B. subtilis* in response to cell growth. The QS system was applied to engineer *Bacillus subtilis* which has the synthesis modules of MK-7, resulting in the BS20 strain. The BS20 strain produced 360 mg/L of MK-7, which was a 40-fold improvement [95]. Recently, Cui et al. (2020) used the *B. subtilis* BS20 strain to construct another engineered strain [97]. Since the authors confirmed that the genes related to cell membrane components could affect the MK-7 production, they overexpressed signal transduction protein (TatAD-CD) and methyl phenol cytochrome c reductase (QcrA-C) in the BS20 strain. The resulting BS20-QT strain produced 310 mg/L of MK-7 [97]. Another study reported the increased production of MK-4 in *B. subtilis* using the QS system [96]. The authors developed the PhrQ-RapQ-ComA QS system and introduced it into the recombinant *B. subtilis*, overexpressing the *ispH*, *crtE*, and *menA* for dynamic control of MK-4 biosynthesis by the QS system. The resulting BC04 strain produced 217 mg/L of MK-4 in a 3L fermenter [96]. These results suggested that cell growth and efficient synthesis of the vitamin K in *B. subtilis* can be dynamically balanced by the QS system, resulting in increased vitamin K production. The highest MK-7 titer by the engineered *Bacillus* was 360 mg/L in 82 h of fermentation, which is higher than the maximum titer by the wild-type *Bacillus* (226 mg/L in 100 h) [95]. These results suggested that the titer of MKs can be further increased by strain engineering.

### 3.2. Engineered Lactic Acid Bacteria

*Lactococcus lactis* is known to produce a range of menaquinone subtypes. Additionally, *L. lactis* has been used as a microbial cell factory platform of LAB. The biosynthesis pathway of menaquinone in *L. lactis* was categorized into four pathways: the mevalonate pathway, the polyprenyl pathway, the shikimate pathway, and the menaquinone pathway [98]. To increase menaquinone production in *L. lactis*, Bøe et al. (2020) overexpressed key enzymes of the menaquinone biosynthetic pathway in *L. lactis***,** including mevalonate kinase (*mvk*)*,* prenyl diphosphate synthases (*preA*), isochorismate synthase (*menF*), and DHNA polyprenyltransferase (*menA*) (Figure 2) [98]. Combined expression of *mvk*, *preA*, and *menA* in *L. lactis* subsp. *cremoris* NZ9000 produced 680 nmol/L of total menaquinones (MK-7, MK-8, and MK-9). When the resulting strain was further utilized for milk fermentation, the menaquinone content (MK-7 to MK-9) in the fermented milk was around 700 nmol/L, which was a 3-fold increase compared to that fermented by the wild type strain (Table 2) [98].

Genetic engineering of LAB strains for vitamin K production has not been widely attempted due to the lack of toolbox and difficulty in strain engineering when compared to other host strains such as *Bacillus* and *E. coli*. Many efforts have been made to develop genetic engineering tools for LAB, including various expression systems and secretion systems [103,104]. CRISPR-Cas systems for LAB have been also developed for *Limosilactobacillus reuteri* (formerly known as *Lactobacillus reuteri*) [105], *Lactobacillus gasseri* [106], *Leuconostoc citreum* [107], *Bifidobacteria* [108], and other strains of LAB [109]. Recent advances in genetic engineering toolbox can facilitate vitamin K production by LAB through rational engineering.

### 3.3. Other Engineered Microorganisms

*E. coli* has been considered as a microbial cell factory platform because of its well-established engineering tool and fast-growing properties [110,111,112]. *E. coli* can synthesize ubiquinone-8 (Q-8) under aerobic conditions and menaquinone-8 (MK-8) under anaerobic conditions [113,114,115]. Kong et al. (2011) tried to increase the production of OPP and chorismite, which are precursors for the tail and head structures of menaquinone, respectively [99]. To select the suitable host for MK-8 production, six strains of *E. coli* (JM109, Top10, DH5a, MG1655, SURE, and MDS42) were evaluated for MK-8 and Q-8 production. The JM109 strain showing the highest content of 55–57 mg MK-8/g WCW (wet cell weight) was selected as a host strain for further metabolic engineering. Overexpression of endogenous IspA, DXR, or IDI increased MK-8 production up to 2-fold. Since MK-8 and Q-8 share chorismate as a precursor for their head polar part, the authors deleted UbiA and UbiC that were involved in the synthesis of Q-8. The Q-8-deficient ubiCA mutant enhanced MK-8 content by 30% compared to wild-type *E. coli*. When MenA was further overexpressed, the MK-8 content was enhanced by 5-fold (290 mg MK-8/g WCW) when compared to the wild-type *E. coli* (Table 2) [99].

*E. coli* can be used to produce MK-7, even if *E. coli* mainly produces Q-8, MK-8, and demethylmenaquinone-8 (DMK-8) [116]. Recently, Gao et al. (2021) produced MK-7 by *E. coli* through systematic metabolic engineering [100]. The authors categorized the MK-7 biosynthesis pathway of *E. coli* into three modules: the MVA pathway, the DHNA pathway, and the MK-7 pathway, and optimized each pathway to enhance MK-7 production. They first overexpressed the isopentenyl diphosphate isomerase (Idi) enzyme from different species, *Saccharomyces cerevisiae, Populus trichocarpa*, and *E. coli* K12, which are involved in the supply of IPP/DMAPP precursor for the biosynthesis of MK-7. Next, they overexpressed the endogenous MenA and exogenous UbiE together with the heptaprenyl pyrophosphate synthase (HepPPS) from *B. subtilis* which enhanced MK-7 production up to 70 μM [100]. The HepPPS is an important key enzyme, allowing MK-7 production in *E. coli* under aerobic conditions. Finally, the DHNA synthetic pathway was enhanced by overexpression of the endogenous MenFDHBCE gene cluster, resulting in 157 μM of MK-7. Membrane engineering was further employed to increase MK-7, which led to 200 μM (129 mg/L) MK-7 in a shake flask experiment. The authors conducted a fed-batch fermentation in a minimal medium using glucose as a substrate and achieved the highest titer of MK-7 ever reported up to 1.35 g/L [100]. The engineered *E. coli* outperformed *Bacillus* and LAB by producing 1.35 g/L of MK-7 during 52 h fermentation with increased titer and productivity when compared to other strains [100]. This result suggested that the microbial cell factory platform can be engineered and used for overproducing vitamin K.

Liu et al. (2017) took a different approach to improve menaquinone synthesis by the engineering of the state of cell membrane in *E. coli* [101]. The authors investigated the role of the cell membrane. To verify whether the state of cell membrane can enhance MK synthesis, they constructed two engineered strains, *E. coli* DH5α FatB and DH5α FatA, which harbored FatB type acyl-ACP thioesterase from *Umbellularia californica* and the FatA type thioesterase from Sunflower, respectively. Since non-ionic surfactant-POE and plant oil-cedar wood oil (CWO) are known to increase extracellular secretion and intracellular synthesis of MK, the effects of these two chemicals on cell morphology and MK production of *E. coli* were investigated. As a result, *E. coli* DH5α FatB exhibited strong MK secretion ability, resulting in 10.71 ± 0.19 mg/L of extracellular MK [101].

Menaquinone-producing non-model microorganism, *F. meningosepticum,* was also engineered to enhance the menaquinone production by random mutagenesis. Tani et al. (2018) used NTG and HNA to construct *F. meningosepticum* mutants overproducing menaquinone [102]. Using glycerol as a substrate, the resulting mutant strain *F. meningosepticum* HNA 350-22 produced 23 mg/L of menaquinone, while the wild-type *F. meningosepticum* IFO 12535 strain produced 14.1 mg/L of menaquinone [102].

## 4. Conclusions

Many studies for production of vitamin K using microorganisms have been conducted. *Bacillus* spp. are the most intensively studied microorganisms for vitamin K production. Many wild-type strains of *Bacillus* producing vitamin K have been isolated from soybean fermented foods, such as *Natto* and *Cheonggukjang*. Additionally, various LAB strains naturally producing different forms of vitamin K have been isolated and utilized for the manufacturing of vitamin K-enriched dairy products. Other than *Bacillus* and LAB, few microorganisms have been studied and characterized for vitamin K production. Most of them were isolated from gut microbiota, which contributes to vitamin K production in the human body. Several microorganisms producing vitamin K are not GRAS microorganisms, which limits the registration and commercialization of fermented products. To utilize the microbial-derived vitamin K as a food supplement, microbial production hosts are required to be safe and registered as GRAS microorganisms. The GRAS wild-type microorganisms are safe to use, but typically exhibit low yields of vitamin K. To improve the vitamin K production by the wild-type microorganisms, optimization of fermentation conditions and medium composition were mainly employed. Additionally, various chemical and physical methods, such as surfactant and ultrasound, have been used for increasing the biosynthesis and secretion of fat-soluble vitamin K out of the microbial cells.Random mutagenesis of the wild-type microorganisms can be used to increase vitamin K production, and vitamin K derived from the mutant strain is okay to use as a food ingredient. However, random mutagenesis strategy requires much time and effort to screen and select the mutants, which increased vitamin K production when compared to rational engineering. High-throughput screening methods may be utilized to accelerate the selection of vitamin K-overproducing strains from the mutant library.Microbial vitamin K production has also been improved by rational strain engineering strategies. These include overexpression of rate-limiting enzymes, deletion of competitive pathways, and fine-tuning of gene expression by modular system and quorum-sensing system. Although *Bacillus*, LAB, and other microbial cell factory platform strains have been engineered for overproduction of vitamin K, the current titers are still not high enough to support the commercialization of microbial-derived vitamin K by fermentation. The titer, yield, and productivity of vitamin K by microorganisms should be further improved by advanced genetic and fermentation technologies [117]. Systems metabolic engineering, which integrates various engineering tools of systems biology, synthetic biology, and evolutionary engineering, can facilitate the development of vitamin K-hyperproducing strains [118]. Moreover, in silico metabolic modeling and machine learning may help to develop the vitamin K-producing microbial hosts as industrially competitive [119].A recent trend suggests that most consumers prefer to take “natural” products than “synthetic” products. The microbial derived-vitamin K as a natural food supplement can satisfy the market demand and sustainable development goals. Even if the engineered host strains are considered as genetically modified organisms (GMO), recent advances in genome editing tools such as CRISPR/Cas9 system allow us to construct marker-free recombinant strains, facilitating safe and sustainable production of food ingredients [120].

## Figures and Tables

**Figure 1 microorganisms-10-00554-f001:**
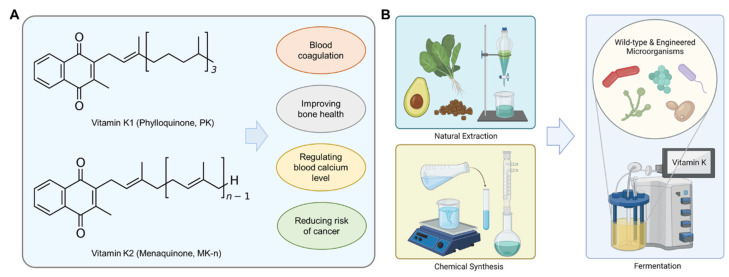
Vitamin K production using microorganisms. (**A**) Chemical structures of vitamin K and its biological functions, and (**B**) Methods of vitamin K production including natural extraction, chemical synthesis, and fermentation.

**Figure 2 microorganisms-10-00554-f002:**
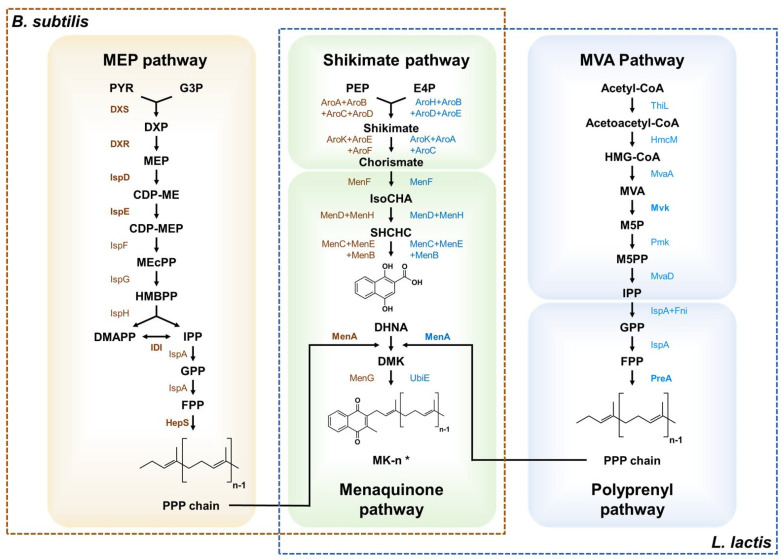
Biosynthetic pathway of menaquinone in *Bacillus subtilis* and *Lactococcus lactis* covered in this review. Metabolic pathways engineered in microorganisms for production of vitamin K were displayed. The prenyl chain structures are synthesized by the MEP pathway in *B. subtilis* and by the MVA pathway in *L. lactis* subsp*. cremoris*. The naphthoquinone ring structure is synthesized by the shikimate pathway and connected to side chains by the menaquinone pathway. The brown square with a dotted line refers to the pathway of menaquinone-7 synthesizing in *B. subtilis*. The blue square with a dotted line refers to the pathway of menaquinones synthesizing in *L. lactis* subsp*. cremoris*. The enzymes in bold were overexpressed to increase vitamin K production in the previous studies. PYR, pyruvate; G3P, glycerol-3-phosphate; DXP, 1-deoxyxylulose-5-phosphate; MEP, methyl-erythritol-4-diphosphate; CDP-ME, 4-(cytidine 5′-diphospho)-2-C-methylerythritol; CDP-MEP, 2-phospho-4-(cytidine 5′-diphospho)-2-C-methylerythritol; MECPP, HMBPP,2-C-Methylerythritol 2,4-cyclodiphosphate; HMBPP,1-hydroxy-2-methyl-2-butenyl 4-diphosphate; DMAPP, dimethylallyl diphosphate; IPP, isopentenyl diphosphate; GPP, geranyl diphosphate; FPP, farnesyl diphosphate; HPP, heptaprenyl diphosphate; PPP, polyprenyl diphosphate; PEP, phosphoenolpyruvate; E4P, erythrose-4-phosphate; IsoCHA, isochorismate; SHCHC, 2-succinyl-6-hydroxy-2,4-cyclohexadiene-1-carboxylate; DHNA, 1,4-dihydroxy-2-naphthoate; DMK, 2-demethylmeaquinone; MK-n, menaquinone-n; MVA, Mevalonate; M5P, Mevalonate 5-phosphate ;M5PP, 5-Diphosphomevalonate; [MEP pathway enzymes] DXS, 1-Deoxy-D-xylulose-5-phosphate synthase; DXR, 1-Deoxy-D-xylulose-5-phosphate reductoisomerase; IspD, 2-C-methyl-D-erythritol 4-phosphate cytidylyltransferase; IspE, 4-Diphosphocytidyl-2-C-methyl-D-erythritol kinase; IspF, 2-C-Methyl-D-erythritol 2,4-cyclodiphosphate synthase; IspG, 4-hydroxy-3-methylbut-2-en-1-yl diphosphate synthase; IspH, 4-Hydroxyl-3-methylbut-2-enyl diphosphate reductase; IspA, Farnesyl diphosphate synthase; HepS, Heptaprenyl diphosphatesynthase component; [Shikimate pathway enzymes (*B. subtilis*)] AroA, 3-Deoxy-7-phosphoheptulonate synthase; AroB, 3-Dehydroquinate synthase; AroC, 3-Dehydroquinate dehydratase; AroD, Shikimate dehydrogenase; AroK, Shikimate kinase; AroE, 3-Phosphoshikimate 1-carboxyvinyltransferase; AroF, Chorismate synthase; [Shikimate pathway enzymes (*L. lactis*)] AroH, Phospho-2-dehydro-3-deoxyheptonate aldolase; AroB, 3-Dehydroquinate synthase; AroD, 3-Dehydroquinate dehydratase; AroE, Shikimate dehydrogenase; AroK, Shikimate kinase; AroA, 3-Phosphoshikimate 1-carboxyvinyltransferase; AroC, Chorismate synthase; [Menaquinone pathway enzymes] MenF, isochorismate synthase; MenD, 2-Succinyl-5-enolpyruvyl-6-hydroxy-3-cyclohexene-1-carboxylate synthase; MenH, Demethylmenaquinone methyltransferase; MenC, O-Succinylbenzoate synthase; MenE, O-Succinylbenzoate-CoA ligase; MenB, 1,4-Dihydroxy-2-naphthoyl-CoA synthase; MenA, 1,4-Dihydroxy-2-naphthoate heptaprenyltransferase; MenG, Demethylmenaquinone methyltransferase; UbiE, Demethylmenaquinone methyltransferase; [Mevalonate pathway enzymes] ThiL, 3-Ketoacyl-CoA thiolase; HmcM, Hydroxymethylglutaryl-CoA synthase; MvaA, Hydroxymethylglutaryl-CoA reductase; Mvk, Mevalonate kinase; Pmk, Phosphomevalonate kinase; MvaD, Diphosphomevalonate decarboxylase; IspA, Geranyltranstransferase; Fni, Isopentenyl-diphosphate delta-isomerase; PreA, Prenyl diphosphate synthase.

**Table 1 microorganisms-10-00554-t001:** Production of vitamin K2 using wild-type microorganisms.

Class	Strain	Strategy	Carbon Source	Menaquinone Type	Fermentation Time	Titer	Extraction	Reference
*Bacillus* spp.	*Bacillus subtilis natto*	Optimization of medium and fermentation condition	Glycerol	MK-7	6 days	62.32 mg/L	By aqueous medium	[49]
	*Bacillus subtilis natto*	Carbon source addition during fermentation	Glycerol	MK-7	6 days	68.6 mg/L	By aqueous medium	[50]
	*Bacillus subtilis natto*	Fed-batch fermentation	Glycerol	MK-7	6 days	86.48 mg/L	By aqueous medium	[51]
	*Bacillus subtilis natto*	Optimization of stirrer speed and aeration rate	Glycerol	MK-7	100 h	226 mg/L	By aqueous medium	[52]
	*Bacillus natto* R127	Optimization of medium Supplementation of surfactant	Glycerol	MK-7	24 h	40.96 mg/L	By aqueous medium	[53]
	*Bacillus subtilis natto* F2	Static fermentation	Glycerol	MK-7	96 h	35.5 mg/L	By fermentation broth	[54]
	*Bacillus subtilis natto* (NF1)	Biofilm reactors (Plastic composite support) Optimization of medium	Glucose	MK-7	144 h	20.5 mg/L	By aqueous medium	[55]
	*Bacillus subtilis natto* (NF1)	Biofilm reactors (Plastic composite support) Optimization of fermentation condition	Glucose	MK-7	144 h	18.45 mg/L	By aqueous medium	[56]
	*Bacillus subtilis natto* (NF1)	Biofilm reactors (Plastic composite support) Fed-batch fermentation	Glucose	MK-7	288 h	28.7 mg/L	By aqueous medium	[57]
	*Bacillus subtilis natto* (NF1)	Biofilm reactors	Glycerol	MK-7	144 h	12.09 mg/L	By aqueous medium	[58]
	*Bacillus subtilis natto* (NF1)	Biofilm reactors (Plastic composite support) Optimization of medium	Glycerol	MK-7	144 h	14.7 mg/L	By aqueous medium	[59]
	*Bacillus subtilis natto* (NF1)	Fermentation in bottle Optimization of medium Static fermentation	Glycerol	MK-7	96 h	32.5 mg/L	By aqueous medium	[60]
			Glucose	MK-7	96 h	14.6 mg/L	By aqueous medium	
	*Bacillus subtilis natto*	Optimization of aeration and agitation	Milk medium	MK-7	72 h	3.54 mg/L	By aqueous medium	[61]
	*Bacillus subtilis* NCIM 2708	Optimization of medium	Glycerol, mannitol	MK-7	24 h	39.039 mg/g	By soybean samples	[62]
	*Bacillus subtilis* ATCC 6633	Iron oxide nanoparticles coated	Glycerol	MK-7	108 h	37.36 mg/L	By aqueous medium	[63]
	*Bacillus amyloliquefaciens* KCTC 11712BP	Optimization of medium and fermentation condition	Glycerol	MK-4	36 h	0.76 mg/g	Fermented *Cheonggukjang*	[64]
			Glycerol	MK-7	36 h	11.71 mg/g	Fermented *Cheonggukjang*	
	*Bacillus velezensis* ND	Liquid-state fermentation	Glycerol	MK-7	168 h	52.9 mg/L	By aqueous medium	[65]
		Biofilm-based fermentation	Glycerol	MK-7	144 h	73.3 mg/L	By fermentation broth	
		Solid-state fermentation	Glycerol	MK-7	96 h	150.02 mg/Kg	By fermentation broth	
Lactic acid bacteria	*Lactococcus lactis* ssp. *cremoris* YIT 2011	Batch fermentation	Glucose	MK-7, MK-8, MK-9	48 h	534 nmol/L	By cell using chloroform and methanol	[66]
	*Lactococcus lactis* ssp. *lactis* YIT 2027	Batch fermentation	Glucose	MK-8, MK-9, MK-10	48 h	717 nmol/L	By cell using chloroform and methanol	
	*Leuconostoc lactis* YIT 3001	Soymilk fermentation	Soy milk	MK-7, MK-8, MK-9, MK-10	48 h	2.60 nmol/L	By soymilk culture using chloroform and methanol	
	*Lactobacillus fermentum* LC272	Batch fermentation	Glucose	MK-4	48 h	0.18 mg/L	By cell using hexane and methanol	[67]
	*Lactococcus lactis* ssp. *cremoris* MG1363	Optimization of temperature, carbon source, aeration, and mode of energy metabolism	Trehalose	MK-5, MK-6, MK-7, MK-8, MK-9, MK-10	48 h	5.2-fold increase compared to the control (90 nmol/L medium)	By cell using hexane and 2-propanol	[68]
Others	*Flavobacterium meningosepticum*	Changing extraction solvent	Glycerol	MK-4, MK-5, MK-6	6 days	1.88 mg/g DCW	By cell using methanol	[69]
	*Flavobacterium sp. M1-14*	Using different surfactant with ultrasound	Glycerol	MK	9 days	30.03 mg/L	In aqueous medium	[70]
	*Enterobacter agglomerans*	Isolated from neonatal fecal flora	Tryptone soy broth powder	MK-4	72 h	ND	By cell using methanol and chloroform	[71]
	*Serratia marcescens*							
	*Enterococcus faecium*							

ND: not determined; DCW: dry cell weight.

**Table 2 microorganisms-10-00554-t002:** Production of vitamin K2 using engineered microorganisms.

Class	Strain	Strategy	Carbon Source	Menaquinone Type	Fermentation Time	Titer	Extraction	Reference
*Bacillus* spp.	*Bacillus subtilis natto* OUV23481	UV and analog resistance (HNA, pFP, mFP, β-TA)	Sucrose	MK-7	16 h	3438 μg/100 g	By Natto	[85]
	*Bacillus subtilis* D200-41	Strain mutation (DPA) media optimization	Glycerol	MK-7	6 days	60 mg/L	By aqueous medium	[83]
	*B. subtilis*(natto)-P15-11-1	Strain mutation (NTG, HNA and N^+^ ion-beam) media optimization	Glycerol	MK-7	70 h	3.593 mg/L	By cell using n-hexane	[84]
	*B. subtilis*(natto)-P15-11-1	Strain mutation and media optimization	Lactose	MK-7	144 h	91.25 mg/L	By aqueous medium	[86]
	*Bacillus subtilis*	Strain mutation (1-naphthol and Tween 80)	Glycerol	MK-7	24 h	14.4 μg/mL	By cell and aqueous medium	[87]
	*Bacillus licheniformis*	Strain mutation (kanamycin and shikimate)	Glucose	MK-7	1 h	0.3 nmol/mL	By cell using acetone	[88]
	*Bacillus amyloliquefaciens H.β.D.R.-5*	Strain mutation (HNA, DPA and β-TA)	Corn starch hydrolysates	MK-7	6 days	61.3 mg/L	By aqueous medium	[89]
	*Bacillus amyloliquefaciens* *MK50-36*	Laboratory evolution at 50 °C	Corn starch	MK-7	144 h	57 mg/L	By aqueous medium	[90]
	*Bacillus subtilis*	Strain mutation (1-naphthol and Tween 80)	Glycerol	MK-7	24 h	14.4 μg/mL	By cell and aqueous medium	[87]
	*Bacillus amyloliquefaciens* Y-2	Metabolic engineering (overexpression of *hepS*)	Glucose	MK-7	24 h	273 μg/g DCW	By cell using n-hexane and 2-propanol mixture	[91]
	*Bacillus subtilis*	Metabolic engineering (overexpression of *Dxs*, *Dxr*, *Idi*, and *MenA*)	Glycerol	MK-7	6 days	50 mg/L	By cell and aqueous medium	[92]
	*Bacillus subtilis MK3-MEP123-Gly2-ΔdhbB*	Metabolic engineering (overexpression of *menA, dxs, dxr, yacM, yacN,* and *glpD,* deletion of *dhbB*	Glycerol	MK-7	120 h	69.5 mg/L	By aqueous medium	[93]
	*Bacillus subtilis* BY23	Metabolic engineering (overexpression of *menA, menG, crtE, dxs, dxr,* and *ispD-ispF,* deletion of *hepT)* and introduction of MVA pathway	Glucose, glycerol	MK-4	144 h	145 mg/L	By cell and aqueous medium	[94]
	*Bacillus subtilis*	Metabolic engineering (introduction of synthesis modules of MK-7) and Rap60-Spo0A quorum sensing system	Glucose	MK-7	6 days	360 mg/L	By cell and aqueous medium	[95]
	*Bacillus subtilis*	Metabolic engineering (overexpressing *ispH, crtE* and *menA*) and PhrQ-RapQ-ComA quorum sensing system	Glucose	MK-4	82 h	217 mg/L	By cell and aqueous medium	[96]
	*Bacillus subtilis* BS20-QT	Metabolic engineering (overexpression of TatAD-CD, QcrA-C)	Glucose, sucrose	MK-7	4 days	310 mg/L	By cell and aqueous medium	[97]
Lactic acid bacteria	*Lactococcus lactis* ssp. *cremoris* MG1363	Metabolic engineering (Overexpression of *mvk*, *preA*, *menA*)	Glucose	MK-7, MK-8, MK-9	Overnight	680 nmol/L	By cell using heptane and 2-propanol	[98]
Others	*Escherichia coli* JM 109	Metabolic engineering (Deletion of ubiC, ubiA, overexpression of MenA, MenD)	Glycerol	MK-8	28 h	290 mg MK-8/g WCW	By cell using chloroform and methanol	[99]
	*Escherichia coli* MK17	Metabolic engineering (Overexpression of Idi, Metk, MenF MenA from *E. coli*, HepPPS, UbiE, from *B. subtilis*)	Glucose	MK-7	52 h	1350 mg/L	By cell using hexane and propanol	[100]
	*E. coli* DH5α Fat B	Metabolic engineering (Overexpression of FatB from *Umbellularia californica*)	Glycerol	MK	120 h	10.71 ± 0.19 mg/L	By cell using methanol	[101]
	*Flavobacterium meningosepticum*	Mutagenesis (NTG, HNA)	Glycerol	MK	72 h	34 mg/L 5.5 mg/g DCW	By cell using acetone and ethyl ether	[102]

HNA: 1-hydroxy-2-naphthoic acid; pFP: p-fluoro-D,L-phenylalanine; mFP: m-fluoro-D,L-phenylalanine; β-TA: β-2-thienylalanine; DPA: diphenylamine; NTG: N-methyl-N′-nitro-N-nitrosoguanidine.

## Data Availability

Not applicable.

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
