# Peer review of "Production of Vitamin K by Wild-Type and Engineered Microorganisms"

_microorganisms, 2022, doi:10.3390/microorganisms10030554_

Round 1
Reviewer 1 Report
The manuscript by Kang et al provides a thorough review of production of vitamin K by wild-type microorganisms and engineered microorganisms with details of strain engineering and fermentation strategies. Vitamin K plays an important role in blood clotting and bone health in human. This review is useful and timely. Overall the manuscript is well written and it is recommended for publication after minor modifications.
1. For some processes, long-time fermentation was required for vitamin K production. It would be appropriate for the authors to provide the fermentation time required for vitamin K production in Table 1 and 2. It would be useful to compare the titer and productivity of products synthesized by different methods. This would allow readers to understand and appreciate the relative merits (and demerits) of all approaches.
2. Figure 2, HPP was demonstrated in “the prenyl chain structures synthesized by the MEP pathway in subtilis”. HPP was for MK-7 synthesis. However, production of MK-4 and MK-7 in B. subtilis was covered in the section. It would be appropriate to provide the prenyl chain structures of menaquinones synthesized in B. subtilis.
Reviewer 2 Report
This study has done a magnificent job of gathering almost all of the literature attempted so far at improving VitK2 fermentation under one roof! Authors have classified and laid out the let review very well. As such, this work will add significant scientific richness to the field of VitK2 fermentation and therefore has significant scientific merit.
The manuscript reads flawlessly with almost no grammatical or even punctual errors.
The tables, graphs and figures are looking great as well.
My only request of the authors is to enrich the pool of literature even further so that the work stands perfectly for the years to come:
Introduction: The authors have actually missed one of the major benefits of vitamin K2 which is preventing CVD:
- https://www.nutraceuticalbusinessreview.com/news/article_page/Vitamin_K2_MK-7_and_Cardiovascular_Calcification/149814
- -https://www.mdpi.com/2072-6643/12/2/583
- https://academic.oup.com/ajcn/article/110/4/883/5544545?login=true
- https://heart.bmj.com/content/105/12/938.abstract?casa_token=-IFHGo2cj70AAAAA:quwjKrfH47-mP85yZI5-Zl3z8Rp4kZVd_jJu-1w5MSZUyZv0hh9upxnMrjxR2Pk247V4IMDeFZ1a
- https://link.springer.com/article/10.1007/s11886-020-1270-1
I suggest an overhaul on the benefits of VitK2 in the intro part and include a segment on CVD mitigations with the many literature on this topic.
Introduction, Line66: Here, authors should also point out that chemically synthesized VitK2 variants even though they are cheaper and purer of Trans isomer (which has the bioactivity in human metabolism) are not considered GRAS:
- https://link.springer.com/article/10.1007/s00253-020-10409-1
- https://www.sciencedirect.com/science/article/pii/S0308814617316308?casa_token=InVg9cPvR_oAAAAA:dF4KKSsBKqHgAt4brWjhWCakRW4rtTLXMgqx_i0GiyhJhMhcQVdZniGbCB9b7cDSKkUkgvIXsD4
The tables look very nice, yet can be even more comprehensive:
Table 1:
- Wang et al. co-produced nattokinase and MK-7 in subtilis vars:
https://link.springer.com/article/10.1007/s13213-018-1372-9
- Mahdinia et al. have also secreted extracellular MK-7 in various subtilis media compositions: https://akjournals.com/view/journals/066/48/4/article-p405.xml
Table 2:
- Goodman et al. have been able to secrete MK-7 in licheniformis as well:
https://journals.asm.org/doi/abs/10.1128/jb.125.1.282-289.1976
- Wu and Ann have done it in amyloliquefaciens as well:
https://link.springer.com/article/10.1007/s10068-011-0219-y
- Puri et al. also have developed a mutant subtilis:
https://www.researchgate.net/profile/Alka-Puri/publication/282745035_Influence_of_physical_chemical_and_inducer_treatments_on_menaquinone-7_biosynthesis_by_Bacillus_subtilis_MTCC_2756/links/5b561686aca27217ffb6c1b0/Influence-of-physical-chemical-and-inducer-treatments-on-menaquinone-7-biosynthesis-by-Bacillus-subtilis-MTCC-2756.pdf
Production of Vitamin K by wild-type microorganisms, Line 119:
It is helpful here to note that MK-7 secretion in the wildtype B. subtilis natto follows neither a primary nor a secondary but actually a mixed-type metabolite pattern:
- https://www.sciencedirect.com/science/article/abs/pii/S1878818118303116
Also, biofilm reactors apparently are claimed to provide significantly shorter lag phase periods and also higher yields:
- https://link.springer.com/article/10.1007/s00253-019-09913-w
Production of Vitamin K using engineered microorganisms, Line 260:
Cui et al have also cell-membrane engineered B. subtilis to enhance MK-7 levels which is worth noticing:
https://www.sciencedirect.com/science/article/pii/S2589004220301024
With the above additions, I think the manuscript will be in great shape for publication.
